# Anyon quantum dimensions from an arbitrary ground state wave function

Shang Liu [1] ✉

Realizing topological orders and topological quantum computation is a central task of modern physics. An important but notoriously hard question in this endeavor is how to diagnose topological orders that lack conventional order parameters. A breakthrough in this problem is the discovery of topological entanglement entropy, which can be used to detect nontrivial topological order from a ground state wave function, but is far from enough for fully determining the topological order. In this work, we take a key step further in this direction: We propose a simple entanglement-based protocol for extracting the quantum dimensions of all anyons from a single ground state wave function in two dimensions. The choice of the space manifold and the ground state is arbitrary. This protocol is both validated in the continuum and verified on lattices, and we anticipate it to be realizable in various quantum simulation platforms.

Topologically ordered phases of matter exhibit a number of remarkable properties, such as the existence of fractionalized excitations dubbed anyons, and robust ground state degeneracies on topologically nontrivial spaces[1]. From a practical perspective, they are also promising platforms for fault-tolerant quantum computation[2,3].

Unlike symmetry breaking orders, topological orders (TOs) lack conventional order parameters. They do not even require any symmetry and sometimes support gapped boundaries. Therefore, diagnosing TOs is generically a difficult task. Recent advances in quantum simulating topologically ordered states[4–12] have further highlighted the need for efficient protocols to identify them. A breakthrough in this problem is the discovery of topological entanglement entropy (EE)[13–16]. It is shown that the EE of a disk region in a two-dimensional gapped ground state wave function contains a universal term dubbed the topological EE, from which we can read off the so-called total quantum dimension $\mathcal{D}$ of the system. $\mathcal{D} = 1$ ($\mathcal{D} > 1$) for a trivial (nontrivial) TO, and hence the topological EE can be used for detecting nontrivial TOs. However, $\mathcal{D}$ is still far from fully characterizing a TO. In particular, it can not distinguish abelian and nonabelian TOs which have very different properties and applications. There have been efforts to extract other universal quantities of a TO, such as the chiral central charge either from edge thermal transport[17–19] or the bulk wave function[20–23], the higher central charge[24], and the many-body Chern number[25–27].

However, these quantities again do not distinguish abelian and nonabelian TOs, and vanish for many TOs supporting gapped boundaries.

If we know the quantum dimensions $d_j$ of all anyon types $j$, we will be able to tell apart abelian and nonabelian TOs, because the former have $d_j = 1$ for all $j$, while the latter have $d_j > 1$ for some $j$. Intuitively, $d_j$ is the Hilbert space dimension shared by each type-$j$ anyon in the limit of many anyons. More precisely, let $D_j(M)$ be the degeneracy of a particular anyon configuration with $M$ type-$j$ anyons. Then in the large $M$ limit, $D_j(M)/d_j^M$ is of order $1$[3]. $\mathcal{D}$ is related to $d_j$'s by $\mathcal{D}^2 = \sum_j d_j^2$. $d_j$'s impose nontrivial constraints on the fusion rules of anyons, and if the chiral central charge is known, they also constrain the anyon self-statistics[19]. Therefore, $d_j$'s are important quantitative characterizations of the anyon excitations.

In this paper, we propose a very simple protocol for extracting the quantum dimensions of all anyons from an arbitrary ground state of a TO on an arbitrary space, e.g. a disk. There are other existing methods to extract $d_j$[15,19,28–34] as well. Some of these methods require knowing the operators for creating anyons, which is unlikely in the case of an unknown wave function. Some other methods require particular state(s) on a torus, which is harder to experimentally prepare than states on a planar geometry. The approach of ref. 33 does not need either of these two, but requires accessing some infinite set of density matrices, which is more of conceptual than practical significance. The

[1]Kavli Institute for Theoretical Physics, University of California, Santa Barbara, CA 93106, USA. ✉e-mail: sliu.phys@gmail.com

key outstanding feature of our proposal is that we have avoided the aforementioned requirements.

In the rest of the paper, we will first describe our protocol, then justify it with a field-theoretic approach, and finally test it on lattices using Kitaev's quantum double models[2].

## Results

### Protocol

Consider a two-dimensional topologically ordered system on an arbitrary space manifold with or without a boundary. Let $|\psi\rangle$ be any state with no excitations in a large enough region, say a ground state. We will describe and later justify an efficient protocol for extracting the quantum dimensions of all anyons.

Consider a partition of the space as shown in Fig. 1 in a region with no excitations. $A = \bigcup_{i=1}^{4} A_i$ takes an annulus shape, and $B$ is the rest of the system. Our protocol consists of three steps listed below. Note that we will first describe the protocol as if we are performing an analytical or numerical computation. A possible experimental realization will be given later.

- **Step 1:** Obtain the reduced density matrix $\rho_A := \mathrm{Tr}_B |\psi\rangle\langle\psi|$ for the annulus region $A$.
- **Step 2:** Map $\rho_A$ to a pure state in the doubled Hilbert space: Let $\rho_A = \sum_{ij} M_{ij} |i\rangle\langle j|$ where $\{|i\rangle\}$ is an arbitrary real-space tensor product basis for Region $A$. We define

$$|\rho_A\rangle := \frac{1}{\sqrt{\mathrm{Tr}(\rho_A^2)}} \sum_{i,j} M_{ij} |i\rangle |j\rangle. \tag{1}$$

- **Step 3:** Denote the doubled system by $A \cup A'$, and divide $A'$ as $\bigcup_{i=1}^{4} A_i'$ in the same way as $A$. Compute the Renyi mutual information $I^{(n)}$ (defined later) between $A_1 \cup A_1'$ and $A_3 \cup A_3'$ for several different Renyi indices $n$, and solve the anyon quantum dimensions $d_j$ according to the following formula.

$$I^{(n)}(11', 33') = \frac{1}{n-1} \log\left[ \sum_j \left( \frac{d_j}{\mathcal{D}} \right)^{4-2n} \right], \tag{2}$$

where $ii'$ stands for $A_i \cup A_i'$, and $\mathcal{D} := \sqrt{\sum_j d_j^2}$ is the total quantum dimension.

Here, the Renyi mutual information is defined as usual by $I^{(n)}(X, Y) := S_X^{(n)} + S_Y^{(n)} - S_{X \cup Y}^{(n)}$, where $S_P^{(n)} := (1-n)^{-1} \log \mathrm{Tr}(\rho_P^n)$ is the Renyi entropy.

Intuitively, $|\rho_A\rangle$ is a particular ground state of the TO on the torus obtained by gluing $A$ and $A'$ along their boundaries. We will later carefully justify this picture and determine this special state. Once this is done, Eq. (2) follows from a known result about mutual information on torus[32].

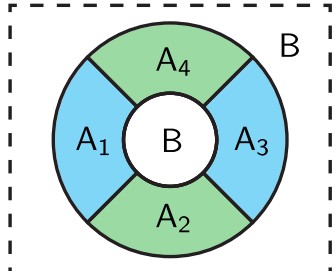

**Fig. 1 | The partition of space used in our protocol.** There is an annulus region $A = \bigcup_{i=1}^{4} A_i$ subdivided into four parts. The remaining uncolored region is $B$.

A few comments are in order. In Step 2, the map from $\rho_A$ to $|\rho_A\rangle$ is basis dependent. If we choose a different real-space tensor product basis, then the new pure state $|\rho_A\rangle_{\mathrm{new}}$ is related to the old one by a local basis rotation in $A'$. This does not affect the entanglement based quantity $I^{(n)}$ that we need. In Step 3, a possible strategy for solving all $d_j$ is as follows: First obtain $I^{(2)}$ which gives the total number of anyon sectors $t$. Then obtain $I^{(n)}$ for more than $t$ number of additional Renyi indices, from which we can uniquely determine all $d_j / \mathcal{D}$. Since we know the smallest quantum dimension is that of the vacuum sector, $d_0 = 1$, we can subsequently find the values of $\mathcal{D}$ and all $d_j$. Note that for an abelian TO where $d_j = 1$ for all $j$, $I^{(n)} = 2 \log \mathcal{D}$ is $n$-independent. Hence, if we are accessible to only a limited number of Renyi indices, although we are not able to obtain all $d_j$, we can still tell whether the TO is abelian or nonabelian. We shall also remark that Renyi EEs for different Renyi indices $n$ have rather different quantum information properties. For example, the strong subadditivity condition holds only for $n = 1$[35]. Our proof of Eq. (2), as we will see later, does not utilize such kind of quantum information property and thus holds for all $n$.

For integer values of $n$, the quantity $I^{(n)}(11', 33')$ proposed above can in principle be experimentally measured. To see this, we need to first understand how to prepare the state $|\rho_A\rangle$ in practice. Let $\{|i\rangle\}$ and $\{|\mu\rangle\}$ be orthonormal bases of $A$ and $B$, respectively. We can write $|\psi\rangle = \sum_{i,\mu} \psi_{i\mu} |i, \mu\rangle$. It follows that $|\rho_A\rangle \propto \sum_{i,j,\mu} \psi_{i\mu} \psi_{j\mu}^* |i\rangle |j\rangle$. Denote by $|\psi^*\rangle = \sum_{i,\mu} \psi_{i\mu}^* |i, \mu\rangle$ the time-reversed copy of $|\psi\rangle$, i.e. the conjugate state in the chosen basis. We observe that $|\rho_A\rangle$ is proportional to $\langle\Psi|(|\psi\rangle \otimes |\psi^*\rangle)$, where $|\Psi\rangle \propto \sum_\lambda |\lambda\rangle|\lambda\rangle$ is a maximally entangled state living in two copies of $B$. This is illustrated using tensor diagrams in Fig. 2. It means that to prepare $|\rho_A\rangle$, we may first prepare the state $|\psi\rangle \otimes |\psi^*\rangle$, and then implement a partial projection onto the state $|\Psi\rangle$ using projective measurements with postselections. If $|\psi\rangle$ can be prepared in a quantum simulation platform using unitary circuits and measurements, then it should be equally easy to prepare the time-reversed copy $|\psi^*\rangle$. Once $|\rho_A\rangle$ can be prepared, one can measure EEs (and therefore the mutual information) for integer Renyi indices $n \geq 2$ using established methods. For example, to measure the $n$-th Renyi EE of a subregion $R$ in a pure state, it suffices to measure the expectation value of the "shift operator" $C_n$. By definition, $C_n$ acts on $n$ copies of the same pure state, and its effect is to cyclically permute the $n$ copies of subregion $R$. We note that both the postselections required for obtaining $|\rho_A\rangle$ and the measurement of EEs require resources that scale exponentially with the system size. Nonetheless, this is not a problem in principle. Since we are dealing with gapped quantum systems with finite correlation lengths, there is no need to go to very large system sizes to get accurate results – We just need the size of each subregion to well exceed the correlation length.

### Continuum approach

In this section, we will give a field-theoretic proof of Eq. (2), assuming the underlying TO to be described by a Chern-Simons (CS) theory[36]. The readers need not be familiar with CS theories, and just need to know that (1) a CS theory is a gauge theory with some compact gauge group $G$, and (2) it is a topological field theory, meaning that the action has no dependence on the spacetime metric and only the spacetime topology matters. As mentioned previously, we require the state $|\psi\rangle$ to have no excitations (zero energy density) in a large enough region. We expect that the reduced density matrix of $|\psi\rangle$ on a disk deep inside this region has no dependence on the boundary condition or excitations far away[31,37,38]. Hence, for simplicity, we assume $|\psi\rangle$ to be the unique ground state of the TO on a sphere. In the CS theory, up to a normalization factor, this state can be prepared by performing the path integral in the interior of the sphere, i.e. on a solid ball, as illustrated in Fig. 3a.

Given the path integral representation of $|\psi\rangle$, we take a mirror image for $\langle\psi|$ (One may check from the CS theory action that taking the complex conjugate of the wave function is equivalent to a mirror reflection of the spacetime manifold.) as shown in Fig. 3b, and then

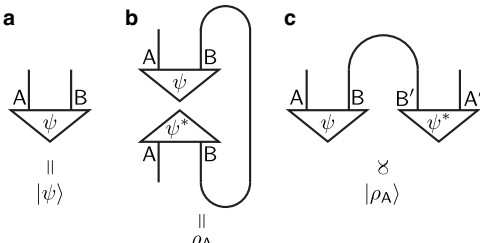

**Fig. 2 | Tensor diagrams for the states used in the protocol.** Panels **a**–**c** represent $|\psi\rangle$, $\rho_A$, and $|\rho_A\rangle$, respectively. Panel **c** also illustrates a practical way of preparing the state $|\rho_A\rangle$.

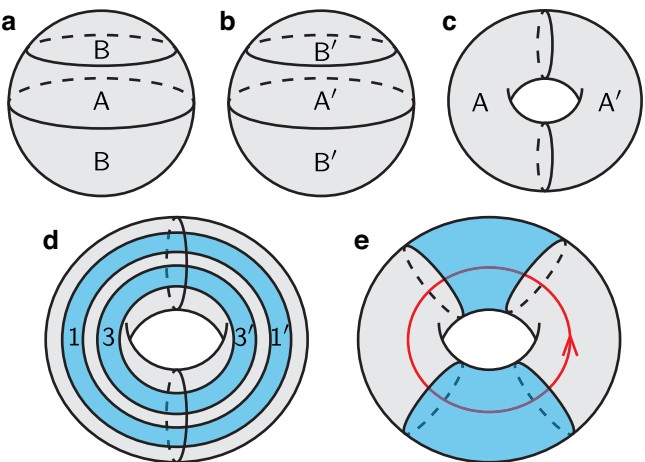

**Fig. 3 | Illustration of the field-theoretic approach. a** The sphere state $|\psi\rangle$ is prepared by doing the path integral on the solid ball as indicated by the gray shade. **b** Path integral for $\langle\psi|$. **c** Path integral for $\rho_A$ or $|\rho_A\rangle$. **d** Regions $A_1 \cup A_1'$ and $A_3 \cup A_3'$. **e** The effect of an $\mathcal{S}$ transformation.

glue together $B$ of $|\psi\rangle$ and $B'$ of $\langle\psi|$ to obtain the path integral for $\rho_A$. The result is shown in Fig. 3c. Up to a normalization, $|\rho_A\rangle$ has the same path integral representation as $\rho_A$, it is therefore a state living on the torus. The Hilbert space on a torus without anyon excitation is multidimensional. An orthonormal basis of the space, denoted by $\{|R_j\rangle\}$, one-to-one corresponds to a finite set of representations $\{R_j\}$ of the gauge group, and also one-to-one corresponds to the anyon types of the TO. The state $|R_j\rangle$ can be prepared by performing the path integral on the solid torus (bagel) with a noncontractible Wilson loop (A Wilson loop is a certain observable defined on an oriented loop and labeled by a representation of the gauge group. It can be regarded as an anyon world line.) carrying the corresponding representation $R_j$ inserted. As shown in Fig. 3c, the path integral for $|\rho_A\rangle$ has no Wilson loop insertion. The state thus corresponds to the trivial representation, or the trivial anyon sector (vacuum).

By keeping track of the subregions of $A$, we observe that $A_1 \cup A_1'$ and $A_3 \cup A_3'$ correspond to two annuli shown in Fig. 3d. We are now supposed to compute the Renyi mutual information between these two regions. To this end, it is convenient to first apply an $\mathcal{S}$ transformation[36], whose effect is shown in Fig. 3e: The two annuli now wind in the perpendicular direction, and a Wilson loop is inserted in the path integral. This new torus state is given by $\sum_j \mathcal{S}_{0j}|R_j\rangle$, where $\mathcal{S}_{0j} = d_j/\mathcal{D}$ are components of the modular $\mathcal{S}$ matrix. The desired mutual information $I^{(n)}$ can now be computed using the replica trick and the surgery method[28,32,36]. In fact, this has been done in Appendix B.4 of ref. 32 (plug in $\psi_a = \mathcal{S}_{0a}$), and the technique is also pedagogically explained in that paper. We arrive at the result in Eq. (2).

As a fixed-point theory, the CS theory only captures the universal terms in EEs. For a generic gapped field theory or lattice model, the EE

of a region also contains nonuniversal terms such as the "area-law" term proportional to the length of region boundary, and terms due to corners or other sharp features which are inevitable on lattices. We need to discuss whether the quantity $I^{(n)}$ we consider contains any nonuniversal term. For a general gapped theory, we expect the picture of Fig. 3d still holds, although the theory is now not topological and depends on the spacetime metric. If we assume that nonuniversal terms in the EEs are made of local contributions (which are insensitive to changes far away) near the partition interfaces[15,16], then we see that all such terms have been canceled in $I^{(n)}$. For example, the boundary of $A_1 \cup A_1'$ contributes the same nonuniversal terms to $S_{11'}^{(n)}$ and $S_{11' \cup 33'}^{(n)}$, and these terms have been canceled in $I^{(n)}(11', 33')$. We note that the locality assumption about nonuniversal terms does not hold in certain systems with the so-called suprious long-range entanglement[39–45], which will not be considered in this work. As one test of the universality of $I^{(n)}$, one can manually add a local bunch of coupled qubits to the state $|\psi\rangle$ at an arbitrary location, and observe that the final result of $I^{(n)}$ has no dependence on the state of these qubits.

### Test on lattices

In addition to the continuum approach, we have also tested our protocol on lattices using Kitaev's quantum double models[2]. This calculation is rather involved, so in the main text, we will only consider the simplest example – that of the toric code model. The most general cases will be discussed in the Supplementary Information.

Given a square lattice with qubits living on the edges (links), the toric code model is defined by the following Hamiltonian.

$$
\begin{aligned}
H_{\mathrm{TC}} &= -\sum_v \begin{array}{c} \mathrm{X} \\ \mathrm{X} \!-\! \underset{v}{\phantom{X}} \!-\! \mathrm{X} \\ \mathrm{X} \end{array} - \sum_f \begin{array}{c} \mathrm{Z} \\ \mathrm{Z}\,\boxed{f}\,\mathrm{Z} \\ \mathrm{Z} \end{array} \\
&\equiv -\sum_v \mathcal{X}_v - \sum_f \mathcal{Z}_f.
\end{aligned}
\tag{3}
$$

The two set of terms in $H_{\mathrm{TC}}$ are usually called star and plaquette terms, respectively. Each star term $\mathcal{X}_v$ (plaquette term $\mathcal{Z}_f$) is the product of all Pauli-$X$ (Pauli-$Z$) operators surrounding a vertex $v$ (face $f$). These terms all commute with each other, and a ground state of $H_{\mathrm{TC}}$ is a simultaneous eigenstate of them with eigenvalue +1. It is not hard to generalize this definition to more general lattices, hence we can put the toric code model on surfaces of different topologies. On a sphere, $H_{\mathrm{TC}}$ has a unique ground states, but on a topologically nontrivial space such as a torus, $H_{\mathrm{TC}}$ has degenerate ground states.

For simplicity, let us try implementing our protocol on the unique ground state $|\Omega\rangle$ on a sphere. For later convenience, we denote by $G$ the abelian group generated by all $\mathcal{X}_v$ and $\mathcal{Z}_f$ operators. $G$ is an example of the so-called stabilizer groups, and elements of $G$ are called stabilizers[46]. Let the total number of qubits be $N$. $G$ can be generated by $N$ number of independent stabilizers $\{s_1, s_2, \cdots, s_N\}$; for example, we can take this set to be all but one $\mathcal{X}_v$ together with all but one $\mathcal{Z}_f$ (since $\prod_v \mathcal{X}_v = \prod_f \mathcal{Z}_f = 1$). The full density matrix $|\Omega\rangle\langle\Omega|$ can be interpreted as the projector onto the one-dimensional eigensubspace of +1 eigenvalue for all stabilizers in $G$. Hence,

$$
|\Omega\rangle\langle\Omega| = \prod_{i=1}^N \left(\frac{1+s_i}{2}\right) = \frac{1}{2^N}\sum_{g\in G} g.
\tag{4}
$$

We take an annulus region $A$ as shown in Fig. 4a. Let $G_A \subset G$ be the subset of stabilizers acting in $A$, we have

$$
\rho_A = \frac{1}{2^{N_A}}\sum_{g\in G_A} g,
\tag{5}
$$

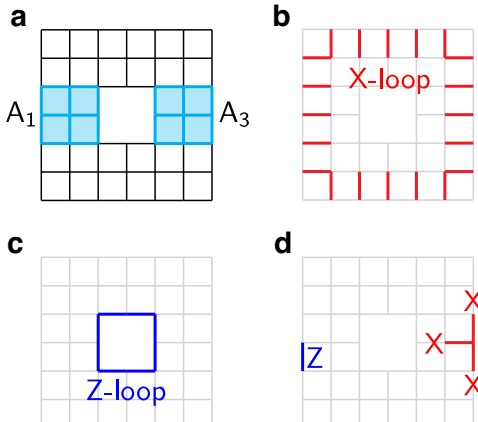

**Fig. 4 | The toric code example. a** The annulus region $A$. Subregions $A_1$ and $A_3$. **b** An $X$-loop operator. **c** A $Z$-loop operator. **d** Examples of boundary operators.

where $N_A$ is the number of qubits in $A$. $G_A$ is generated by all star and plaquette terms in $A$ as well as two loop operators shown in Fig. 4b, c. We denote the two loop operators as $\mathcal{W}_X$ and $\mathcal{W}_Z$, respectively. The $Z$-loop ($X$-loop) operator $\mathcal{W}_Z$ ($\mathcal{W}_X$) is the product of all Pauli-$Z$ (Pauli-$X$) operators along a loop living on edges (dual lattice edges).

We claim that the state $|\rho_A\rangle$ is a ground state of $H_{TC}$ on a torus, where the torus is obtained by taking two copies of $A$ and identifying their corresponding boundary vertices. To prove this claim, we need to verify that $|\rho_A\rangle$ has $+1$ eigenvalue under all star and plaquette terms on the torus. We observe that all these terms have either of the following two forms, where we use $\otimes$ to connect operators acting on the two copies of $A$.

- $\mathcal{O} \otimes 1$ or $1 \otimes \mathcal{O}$, where $\mathcal{O}$ is an $\mathcal{X}_v$ or $\mathcal{Z}_f$ operator acting in $A$.
- $\Delta \otimes \Delta$, where $\Delta$ acts near the boundary of $A$, and two examples of $\Delta$ are given in Fig. 4b.

$|\rho_A\rangle$ satisfies the first set of stabilizers since $\mathcal{O}\rho_A = \rho_A\mathcal{O} = \rho_A$, where we used the fact that $\mathcal{X}_v$ and $\mathcal{Z}_f$ are both real Hermitian operators. $|\rho_A\rangle$ satisfies the second set of stabilizers because all boundary operators $\Delta$ commute with $G_A$ and thus $\Delta\rho_A\Delta = \rho_A$. This finishes the proof of the claim. On a torus, $H_{TC}$ has four degenerate ground states. $|\rho_A\rangle$ can be uniquely determined by specifying two more stabilizers such as $\mathcal{W}_X \otimes 1$ and $\mathcal{W}_Z \otimes 1$.

It remains to compute the Renyi mutual information $I^{(n)}(A_1 \cup A_1', A_3 \cup A_3')$, where $A_1$ and $A_3$ are shown in Fig. 4a. Renyi EE and therefore mutual information can be conveniently computed in the stabilizer formalism[47]. Let $|\psi\rangle$ be an $M$-qubit stabilizer state determined by a stabilizer group $H$. Let $R$ be a subregion with $M_R$ number of qubits and $H_R \subset H$ be the subgroup of stabilizers in $R$. From the reduced density matrix $\rho_R := \text{Tr}_{\bar{R}}(|\psi\rangle\langle\psi|) = 2^{-M_R}\sum_{h\in H_R}h$, one can check that $S_R^{(n)} = (M_R - k_R)\log 2$ where $k_R$ is the number of independent generators of $H_R$, i.e. $|H_R| = 2^{k_R}$. The mutual information between two regions $R_1$ and $R_2$ is therefore given by $(k_{R_1 \cup R_2} - k_{R_1} - k_{R_2})\log 2$, independent on the Renyi index $n$. In our case, $R_1 = A_1 \cup A_1'$ and $R_2 = A_3 \cup A_3'$ are two annuli on the torus. $H_{R_1 \cup R_2}$ has two more generators than $H_{R_1}H_{R_2}$: We can take the first (second) generator as the product of two noncontractible $X$-loop ($Z$-loop) operators in $R_1$ and $R_2$, respectively. We thus find $I^{(n)}(R_1, R_2) = 2\log 2$ for all $n$. This is indeed consistent with the fact that toric code is an abelian TO with $\mathcal{D} = 2$.

For general quantum double models, we find more interesting results of the mutual information, revealing nontrivial quantum dimensions. We refer interested readers to the Supplementary Information for details.

## Discussion

In this work, we have introduced a simple protocol for extracting all anyon quantum dimensions of a two-dimensional TO from an arbitrary ground state wave function. It is both validated in the continuum and verified on lattices. It is interesting to seek generalizations of this protocol for extracting more universal information, such as the fusion rules, $\mathcal{S}$ matrix, and topological spins.

We should mention that this work is partially inspired by ref. 48, which studies the entanglement negativity between two spatial regions in a tripartite topologically ordered state with trisection points (points where the three regions meet). Using some "wormhole" approach, it is found that the negativity contains an order-1 term that can distinguish abelian and nonabelian TOs. However, it is not clear at least to us whether this term is comparable to any universal quantity in generic models. It actually seems hard to extract a universal term from the entanglement negativity with trisection points[34,49,50], and more studies are needed to better understand this issue.

Finally, we note that this work is still not totally satisfactory: Our protocol has only been checked using fixed-point models, either in the continuum or on lattices. Future numerical simulations are needed to further verify this protocol in the presence of perturbations.

## Data availability

This research is analytical; there is no numerical or experimental data. Part of the analytical derivations are provided in the Supplementary Information file.

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

## Acknowledgements

I am grateful to Chao Yin for a previous related collaboration[34], to Yanting Cheng and Pengfei Zhang for feedbacks on the manuscript, and to Wenjie Ji, Yuan-Ming Lu, Nat Tantivasadakarn, Ashvin Vishwanath, and Liujun Zou for helpful discussions. I am supported by the Gordon and Betty Moore Foundation under Grant No. GBMF8690 and the National Science Foundation under Grant No. NSF PHY-1748958.

## Author contributions

This is a single-author paper.

## Competing interests

The author declares no competing interests.
