## [Peer Review File · Nature Communications]

REVIEWER COMMENTS

Reviewer #1 (Remarks to the Author):

In this paper, the author presents a simple protocol to extract the anyonic quantum dimensions d_a from a (topologically ordered) ground state wave function. The protocol consists in three steps. 1) to obtain the reduced density matrix of the wave-function for a (large) annulus subsystem A.

2) To purify it in a doubled Hilbert space.

3) Study the correlations by Renyi mutual information between the two copies in a particular choice of the subpartition of A.

The validity of the formula is proven using a continuum approach and then examples are provided based on the toric code model.

The detection of topological order and its characterization through quantum dimensions is definitely one of the most fascinating topics in quantum many-body theory. The algorithm presented here is elegant and very interesting. However, I have some concerns about the validity of the result. I hope the author can help clarify the following questions.

Question 1. At the very onset, in step 1, the authors says "obtain the reduced density matrix ρ_A ". It is not very clear what the author means. Does the author mean that one prepares the ground state and then can access observables in the region A? Does instead mean that one has in fact measured all of them (that is, has performed a complete tomography) in order to obtain all the matrix elements M_{ij} ? Or the rest of the protocol just needs access to the state?

Question 2. This is related to my previous point. How does one perform the construction of Eq(1)? Again, if one has obtained the data M_{ij} then one can just plug them in a computer, but this means that the number of degrees of freedom in A must be very small as there are $q^{(2n_A)}$ matrix elements M_{ij} if n_A is the number of degrees of freedom in A.

Question 3. In step 3 of the algorithm, one computes the Renyi mutual informations of order n obtaining equation (2), from which one can then extract the quantum dimensions d_a . Now, it is not very clear to me if the derivation of equation (2) follows through for all Renyi indexes n . Indeed, the mutual information of order 1 is kind of special (just like the von Neumann entropy) because of the several forms of subadditivity that are guaranteed for $n=1$. For example, in Journal of Mathematical Physics vol. 56, no. 2, article no. 022205 some of these differences are discussed (see also JHEP 12

(2016) 145 for the possible gaps between different Renyi mutual information quantities). The author should really clarify this point.

Question 4. In Phys. Rev. Lett. 103, 261601 (2009), see Equation (6) a similar equation to equation (2) in this paper is found. However, it is shown that the contribution depending on d_a is non universal - unlike what seems to be implied in the present work. This casts some doubt on the validity of the method. Moreover, in the same paper, the authors show that quantum double models possess a flat entanglement spectrum and as such no universal quantity can depend on different Renyi indexes.

To conclude, I find this paper very interesting and of interest for Nature communications. However, I am not yet convinced of the validity, especially because of perhaps having used some properties of mutual information that are special to $n=1$ for all the Renyi indexes.

I am willing to review again the paper but the author should address all my concerns convincingly.

Reviewer #3 (Remarks to the Author):

Reviewer #4 (Remarks to the Author):

I am writing to present my evaluation and recommendation regarding the manuscript entitled "Anyon Quantum Dimensions from an Arbitrary Ground State Wave Function," authored by Shang Liu. This work introduces an innovative entanglement-based methodology for deducing the quantum dimensions of anyons from a singular two-dimensional ground state wave function. Topological orders and anyons are essential in the realm of topological quantum computation. However, the challenge of diagnosing topological orders persists, primarily due to the absence of conventional order parameters. The advent of topological entanglement entropy has marked a significant advancement in identifying nontrivial topological orders. Nonetheless, this measure alone does not completely characterize topological orders. The manuscript in question addresses this gap through a

protocol that adeptly navigates the limitations of current methods. This protocol could be applied in both experimental and numerical studies. It involves a series of steps, starting with obtaining the reduced density matrix of a chosen area. After acquiring this matrix, the next step is to convert it into a pure state. This pure state is a quantum state that describes a system's properties completely. Finally, the protocol requires calculating the Rényi mutual information between different areas. The Rényi mutual information is a measure used in quantum information theory to quantify the information shared between parts of a system. By performing this calculation, the protocol enables the determination of the quantum dimensions of anyons. Quantum dimensions are critical parameters that characterize anyons, which are quasiparticle types existing in two-dimensional quantum systems and play a vital role in topological quantum computation. The validation of this protocol is meticulously conducted through the application of Chern-Simons field theories and Kitaev's quantum double models. Furthermore, the manuscript furnishes a field-theoretic justification of the proposed protocol and delves into the discussion of nonuniversal terms within the entanglement entropy.

While I endorse the paper for its significant contribution, I do have queries and suggestions, and the manuscript should be revised before publication:

1. About the presentation format of this manuscript, the distinction between known facts in the literature and the new ideas proposed by the author is unclear. For example, the most fundamental formula to obtain the quantum dimensions of anyons is Eq. (2), and the arguments and conclusions follow from this equation straightforwardly. However, this formula is not derived in the main text and is referred to Ref. [28]. I suggest that the author emphasizes his contribution more clearly.

2. In the introductory section of the paper titled "Rényi Mutual Information in Quantum Field Theory, Tensor Networks, and Gravity" by Jonah Kudler-Flam, Laimei Nie, and Akash Vijay (<https://arxiv.org/abs/2308.08600>), an issue is highlighted concerning the conventional definition of Rényi mutual information. The authors note that under this conventional definition, the data processing inequality does not hold, rendering Rényi mutual information an unreliable measure of correlation. This raises concerns regarding the robustness of the anyon quantum dimensions derived in the manuscript, as these dimensions may not represent a meaningful physical quantity. The manuscript's proposal has been verified using exactly solvable models, such as toric codes and Kitaev's quantum double models. However, the applicability and accuracy of this method when applied to non-fixed-point wavefunctions of topological phases remain untested. It is imperative to explore the protocol's performance in scenarios that deviate from ideal conditions. I suggest conducting numerical analyses on wavefunctions that have been modified by unitary operations or even those subjected to minor random disturbances. This would provide a more comprehensive understanding of the protocol's reliability and potential as a tool for studying the quantum dimensions of anyons in more realistic conditions.

3. I have suggestions for related references for the introduction to provide a more comprehensive background.

Extracting Higher Central Charge from a Single Wave
function(<https://doi.org/10.1103/PhysRevLett.132.016602>)

Many-Body Chern Number from Statistical Correlations of Randomized Measurements
(<https://doi.org/10.1103/PhysRevLett.126.050501>)

Extraction of the many-body Chern number from a single wave
function(<https://doi.org/10.1103/PhysRevB.103.075102>)

In light of the above, I advocate for the publication of this manuscript in Nature Communications. The study not only contributes significantly to the field by refining our understanding and diagnostic capabilities of topological orders but also paves the way for further experimental and numerical implementations of the protocol. It stands as a testament to innovative research that can foster advancements in quantum computing.

REVIEWER COMMENTS

Reviewer #1 (Remarks to the Author):

In this paper, the author presents a simple protocol to extract the anyonic quantum dimensions d_a from a (topologically ordered) ground state wave function. The protocol consists in three steps. 1) to obtain the reduced density matrix of the wave-function for a (large) annulus subsystem A.

2) To purify it in a doubled Hilbert space.

3) Study the correlations by Renyi mutual information between the two copies in a particular choice of the subpartition of A.

The validity of the formula is proven using a continuum approach and then examples are provided based on the toric code model.

The detection of topological order and its characterization through quantum dimensions is definitely one of the most fascinating topics in quantum many-body theory. The algorithm presented here is elegant and very interesting. However, I have some concerns about the validity of the result. I hope the author can help clarify the following questions.

Reply: I would like to thank the reviewer for the very positive feedback, and appreciate the concerns raised. I will address these concerns below.

Question 1. At the very onset, in step 1, the authors says "obtain the reduced density matrix ρ_A ". It is not very clear what the author means. Does the author mean that one prepares the ground state and then can access observables in the region A? Does instead mean that one has in fact measured all of them (that is, has performed a complete tomography) in order to obtain all the matrix elements M_{ij} ? Or the rest of the protocol just needs access to the state?

Reply: I am grateful to the reviewer for this good question. In fact, I wrote down these protocol steps as if I was doing an analytical or numerical calculation. I had in mind an experimental

realization, which does not require full tomography, but I did not write it down. To fix this, I have made two modifications:

(1) Before the description of these protocol steps, I added a sentence like this: “Note that we will first describe the protocol as if we are performing an analytical or numerical computation. A possible experimental realization will be given later.”

(2) At the end of this subsection, I added a long paragraph (with a new figure) explaining a possible experimental realization of the protocol in quantum simulation platforms.

These changes have been highlighted with color in the revised manuscript and should be easy to find.

Question 2. This is related to my previous point. How does one perform the construction of Eq(1)? Again, if one has obtained the data M_{ij} then one can just plug them in a computer, but this means that the number of degrees of freedom in A must be very small as there are $q^{(2n_A)}$ matrix elements M_{ij} if n_A is the number of degrees of freedom in A .

Reply: I would like to thank the reviewer for raising this important question. A detailed answer to this question is given in the new paragraph and figure just mentioned. In short, one need to prepare both the original pure state $|\psi\rangle$ and its conjugate state $|\psi^\rangle$ -- which shouldn't be hard if $|\psi\rangle$ is to be prepared in a quantum simulation platform using unitary circuits + measurements. Then, one need to implement a projection operator using measurements + postselections.*

Question 3. In step 3 of the algorithm, one computes the Renyi mutual informations of order n obtaining equation (2), from which one can then extract the quantum dimensions d_a . Now, it is not very clear to me if the derivation of equation (2) follows through for all Renyi indexes n . Indeed, the mutual information of order 1 is kind of special (just like the von Neumann entropy) because of the several forms of subadditivity that are guaranteed for $n=1$. For example, in Journal of Mathematical Physics vol. 56, no. 2, article no. 022205 some of these differences are discussed (see also JHEP 12 (2016) 145 for the possible gaps between different Renyi mutual information quantities). The author should really clarify this point.

Reply: I would like to thank the reviewer for pointing out this possible confusion. Our derivation of Eq.2 does not involve any quantum information property of Renyi EEs (essentially only the definition has been used). Hence, Eq.2 indeed holds for all Renyi indices n . In the revised manuscript, we have added a remark to clarify this point, at the end of the second-last paragraph of the protocol subsection: “We shall also remark that Renyi EEs for different Renyi indices $n\dots$ ”. The paper of Berta et al suggested by the reviewer has also been cited there. I did not cite the JHEP paper since it does not seem to be very directly related (but please do let me know if the reviewer thinks it is necessary to include that one).

Question 4. In Phys. Rev. Lett. 103, 261601 (2009), see Equation (6) a similar equation to equation (2) in this paper is found. However, it is shown that the contribution depending on d_a is non universal - unlike what seems to be implied in the present work. This casts some doubt on the validity of the method. Moreover, in the same paper, the authors show that quantum double models possess a flat entanglement spectrum and as such no universal quantity can depend on different Renyi indexes.

Reply: We thank the review for the question. The paper that the reviewer mentioned is considering a rather different situation from my paper. In their paper, then consider a simply connected region in the conventional Hilbert space. In my paper, I am considering some special region in the doubled Hilbert space, which turns out to be equivalent to some noncontractible region on a torus. Therefore, there is no contradiction between their and my results. I hope this clarifies the confusion.

To conclude, I find this paper very interesting and of interest for Nature communications. However, I am not yet convinced of the validity, especially because of perhaps having used some properties of mutual information that are special to $n=1$ for all the Renyi indexes.

I am willing to review again the paper but the author should address all my concerns convincingly.

Reply: I deeply appreciate the time and effort of the reviewer, and the positive feedback. I hope my responses have clarified all the concerns.

Reviewer #2 (Remarks to the Author):

This paper introduces a novel protocol for extracting the quantum dimensions of anyons using a single ground state wave function in two-dimensional systems. By calculating the n -th Rényi mutual information between two regions within the doubled reduced density matrix, the author simplifies the complex problem of determining quantum dimensions of anyons to a matter of solving linear equations.

The justification for the results is provided through two aspects, the comparison with field-theoretic results in the cited papers and the calculation from exactly solvable models, the toric code model within the main text and the quantum double model in the appendix. We think the derivation of the results is rigorous and solid.

The author also acknowledges the limitation regarding the locality assumption of the nonuniversal terms. It is noted that the result doesn't hold in systems with spurious long-range entanglement.

Reply: I deeply appreciate the time and effort of the reviewers, and the positive feedback.

However, we still have several questions about this paper which are listed below:

- Eq. (2) implies that ground states that support the same total number of anyon sectors have the same second Rényi mutual information between $11'$ and $33'$. Is there an intuitive explanation of it?

Reply: I would like to thank the reviewers for this very interesting question, but unfortunately, I do not have a good intuition for that. In fact, it may be that there just isn't a nice intuition, for the following reason. I feel that Rényi entropy has less information theoretic meaning than von Neumann entropy. For example, the latter satisfies the strong subadditivity condition while the former generically does not. But anyway, I will continue thinking about this question and update my paper if I get an answer.

- In Appendix C 1, the author defines the operator $A_{\omega_i}(h)$ in Eq. (C1). $A_{\omega_i}(h)$ alone is not a projector, only the sum over all the group element $h \in G$ is. Therefore, how can it only change the value g_i but keep other holonomy unchanged?

For example, consider the following lattice.

The holonomy basis (if we don't consider the non-contractable loops) can be written as

$$|\omega_1, \omega_2, \omega_3, \omega_4\rangle = |z_2, z_2 z_1^{-1}, z_2 z_4^{-1}, z_2 z_3^{-1}\rangle. \quad (1)$$

After applying $A_{\omega_1}(h)$ we get

$$A_{\omega_1}(h)|\omega_1, \omega_2, \omega_3, \omega_4\rangle = |hz_2, h z_2 z_1^{-1} h^{-1}, h z_2 z_4^{-1} h^{-1}, h z_2 z_3^{-1} h^{-1}\rangle, \quad (2)$$

which is not in general equal to $|hz_2, z_2 z_1^{-1}, z_2 z_4^{-1}, z_2 z_3^{-1}\rangle$ unless summing over h on both sides.

*Reply: I would like to thank the reviewers for this question, and the time and effort spent. The confusion comes from a small error on the right-hand side of equation (1) above. The holonomy is defined as the product of group elements **in the reversed order**. Therefore, for example, the holonomy from ν_0 to ω_2 is actually $z_1^{-1} z_2$ instead of $z_2 z_1^{-1}$. Then under the action of A_{ω_1} , this holonomy indeed does not change. I am sorry that the definition of holonomy causes confusion; this convention was chosen to be maximally consistent with Kitaev's original paper. To make things better, I have highlighted "in the reversed order" by italic font in the definition of holonomy, and also added a reminder (marked as blue) in the example that follows.*

- Can the author specify the definition of $(1, g_{2K, m > 1})$ in the third and fourth line of Eq. (C28)?

Reply: I thank the reviewers for this question, and sorry again for the confusion. Let me explain it as follows. In Eq.C25, the first entry in the ket is $g_{2K, m} a_{2K, l}^{-1}$. "m" here is an index that can take different values, so if we vary m, this term really represents a list of terms:

$g_{2K,1} a_{2K-I}^{-1}$, $g_{2K,2} a_{2K-I}^{-1}$, $g_{2K,3} a_{2K-I}^{-1}$, etc.

Now, after the change of variables in Eq.C26, this list of terms becomes:

a_{2K-I}^{-1} , $g_{2K,2} a_{2K-I}^{-1}$, $g_{2K,3} a_{2K-I}^{-1}$, etc. (Note the first term has changed)

I don't have a good compact notation to represent this, so I have used $(1, g_{2K,m>1}) a_{2K-I}^{-1}$.

I have added a detailed explanation of this notation in the manuscript (marked as blue).

- The author claims that the protocol they provided has experimental advantages. It would be beneficial if the author could elaborate on the practical implementation of this protocol within existing experimental frameworks. Specifically, a discussion on the feasibility of mapping the reduced density matrix onto the doubled system would provide valuable insight into the protocol's applicability in real-world settings.

Reply: I would like to thank the reviewers for this valuable suggestion. I have added a long paragraph together with a new figure at the end of the protocol subsection to explain the implementation of the protocol in experiments. I hope this is helpful.

In conclusion, this paper is scientifically interesting and the mathematical derivation is solid. However, there remain questions that we hope will be addressed. Specifically, a discussion on the potential for experimental verification of the protocol would greatly enhance the paper's applicability. Should the author address these concerns, we would be inclined to recommend the paper for acceptance.

Reply: I would like to thank the reviews again for the time and effort, and for the positive feedback. I hope my above responses have addressed all the concerns.

Reviewer #3 (Remarks to the Author):

Reply: I would like to thank the reviewer for the time and effort.

Reviewer #4 (Remarks to the Author):

I am writing to present my evaluation and recommendation regarding the manuscript entitled "Anyon Quantum Dimensions from an Arbitrary Ground State Wave Function," authored by Shang Liu. This work introduces an innovative entanglement-based methodology for deducing the quantum dimensions of anyons from a singular two-dimensional ground state wave function. Topological orders and anyons are essential in the realm of topological quantum computation. However, the challenge of diagnosing topological orders persists, primarily due to the absence of conventional order parameters. The advent of topological entanglement entropy has marked a significant advancement in identifying nontrivial topological orders. Nonetheless, this measure alone does not completely characterize topological orders. The manuscript in question addresses this gap through a protocol that adeptly navigates the limitations of current methods. This protocol could be applied in both experimental and numerical studies. It involves a series of steps, starting with obtaining the reduced density matrix of a chosen area. After acquiring this matrix, the next step is to convert it into a pure state. This pure state is a quantum state that describes a system's properties completely. Finally, the protocol requires calculating the Renyi mutual information between different areas. The Renyi mutual information is a measure used in quantum information theory to quantify the information shared between parts of a system. By performing this calculation, the protocol enables the determination of the quantum dimensions of anyons. Quantum dimensions are critical parameters that characterize anyons, which are quasiparticle types existing in two-dimensional quantum systems and play a vital role in topological quantum computation. The validation of this protocol is meticulously conducted through the application of Chern-Simons field theories and Kitaev's quantum double models. Furthermore, the manuscript furnishes a field-theoretic justification of the proposed protocol and delves into the discussion of nonuniversal terms within the entanglement entropy.

Reply: I deeply appreciate the time and effort of the reviewer, as well as the very positive feedback.

While I endorse the paper for its significant contribution, I do have queries and suggestions, and the manuscript should be revised before publication:

1. About the presentation format of this manuscript, the distinction between known facts in the literature and the new ideas proposed by the author is unclear. For example, the most fundamental formula to obtain the quantum dimensions of anyons is Eq. (2), and the arguments and conclusions follow from this equation straightforwardly. However, this formula is not derived in the main text and is referred to Ref. [28]. I suggest that the author emphasizes his contribution more clearly.

Reply: I am grateful to the reviewer for pointing out this issue. I have modified the manuscript accordingly: In the paragraph (starting with “Intuitively, ...”) that follows the one about the protocol, I have added a sentence “... Eq. 2 follows from a known result about mutual information on torus [previous Ref 28].” All modifications of the manuscript have been highlighted with color.

2. In the introductory section of the paper titled “Rényi Mutual Information in Quantum Field Theory, Tensor Networks, and Gravity” by Jonah Kudler-Flam, Laimei Nie, and Akash Vijay (<https://arxiv.org/abs/2308.08600>), an issue is highlighted concerning the conventional definition of Rényi mutual information. The authors note that under this conventional definition, the data processing inequality does not hold, rendering Rényi mutual information an unreliable measure of correlation. This raises concerns regarding the robustness of the anyon quantum dimensions derived in the manuscript, as these dimensions may not represent a meaningful physical quantity. The manuscript's proposal has been verified using exactly solvable models, such as toric codes and Kitaev's quantum double models. However, the applicability and accuracy of this method when applied to non-fixed-point wavefunctions of topological phases remain untested. It is imperative to explore the protocol's performance in scenarios that deviate from ideal conditions. I suggest conducting numerical analyses on wavefunctions that have been modified by unitary operations or even those subjected to minor random disturbances. This would provide a more

comprehensive understanding of the protocol's reliability and potential as a tool for studying the quantum dimensions of anyons in more realistic conditions.

Reply: I would like to thank the reviewer for this suggestion. I have actually considered the possibility of doing numerics before submitting this paper, but found it too challenging. First, the required Hilbert space dimension is beyond the scope of exact diagonalization – the operation of doubling the Hilbert space is a particularly difficult step. Second, I have also consulted two colleagues with numerical expertise (one on DMRG and the other on Quantum Monte Carlo), but both said that this is a highly nontrivial task. Therefore, although in future I will try finding collaborators to check this result with numerical simulation, this task is currently really beyond my own technical ability, and I hope this is understandable to the reviewer. To better acknowledge this incompleteness, I have added a paragraph at the end of the Discussion to emphasize this issue.

3. I have suggestions for related references for the introduction to provide a more comprehensive background.

Extracting Higher Central Charge from a Single Wave

function(<https://doi.org/10.1103/PhysRevLett.132.016602>)

Many-Body Chern Number from Statistical Correlations of Randomized Measurements

(<https://doi.org/10.1103/PhysRevLett.126.050501>)

Extraction of the many-body Chern number from a single wave

function(<https://doi.org/10.1103/PhysRevB.103q.075102>)

Reply: I would like to thank the reviewer for suggesting these references. They have all been added to the Introduction, color highlighted.

In light of the above, I advocate for the publication of this manuscript in Nature Communications. The study not only contributes significantly to the field by refining our understanding and diagnostic capabilities of topological orders but also paves the way for further

experimental and numerical implementations of the protocol. It stands as a testament to innovative research that can foster advancements in quantum computing.

Reply: I would like to thank again for the reviewer's recommendation.

REVIEWERS' COMMENTS

Reviewer #1 (Remarks to the Author):

The author has addressed all my concerns in a convincing way. Moreover, the author has - I believe - addressed thoroughly also the points raised by the other referees. I believe the manuscript should be accepted in nature communications.

Reviewer #2 (Remarks to the Author):

We thank the author for the detailed response of the questions we raised. The new paragraph explaining the feasibility of conducting the protocol in experiment is also constructive. Therefore, we think this paper is ready for publication.

Reviewer #3 (Remarks to the Author):

Reviewer #4 (Remarks to the Author):

Upon thoroughly evaluating the revised manuscript and the author's responses to the initial feedback, I am pleased to express my endorsement for its publication in Nature Communications. The author has shown respectable diligence in revising the manuscript, thoroughly addressing concerns raised and significantly enhancing the clarity and depth of the study. Below, I integrate my specific scientific commendations with observations on the improvements made to the manuscript.

Scientific Commendations:

The manuscript presents an important contribution to quantum computation, specifically studying anyons within two-dimensional quantum systems. Quantum dimensions, as critical parameters characterizing these quasiparticles, are pivotal for advancing topological quantum computation. The author validates the proposed protocol by rigorously applying Chern-Simons field theories and Kitaev's quantum double models. Furthermore, the manuscript provides a nuanced field-theoretic justification of the protocol, engaging deeply with the discourse on nonuniversal terms within the entanglement entropy. Such meticulous treatment of the subject matter not only refines our understanding and diagnostic capabilities of topological orders but also opens avenues for further experimental and numerical exploration of the protocol.

Revisions and Responses:

The author's genuine appreciation of constructive feedback and the subsequent positive revisions significantly elevate the manuscript's quality. The clarification regarding Eq. 2 and the addition of pertinent references demonstrate a commitment to scholarly rigor and transparency.

The detailed explanation of the challenges of numerical simulations, including consultations with numerical experts, highlights the author's conscientious approach to addressing potential methodological limitations. The added paragraph in the Discussion section acknowledging these challenges enhances the manuscript's integrity.

Incorporating suggested references into the Introduction, duly highlighted, and the general responsiveness to all recommendations underscore the author's openness to constructive critique and dedication to the manuscript's improvement.

Conclusion:

Considering this study's substantive scientific contributions, the author's responsiveness to feedback, and the quality of the revisions made, I strongly recommend the manuscript be published in Nature Communications. The work is a significant addition to the literature on quantum computing, offering innovative insights that promise to stimulate further advancements in the field. The meticulous validation of the proposed protocol and the thoughtful discussion of its theoretical underpinnings exemplifies the high caliber of research that Nature Communications champions.

In summary, this manuscript not only significantly contributes to our understanding of topological quantum computation but also serves as a foundation for future explorations in this rapidly evolving field. I advocate for its publication with great enthusiasm, looking forward to its positive impact on the scientific community.

It is great to see that all reviewers are satisfied with the current version of the manuscript. We would like to thank all reviewers and the editor for their effort. There is no new modification to the manuscript except for some formatting adjustments and typo fixes.

Shang